# Variable Optimization of Seaweed Spectral Response Characteristics and Species Identification in Gouqi Island

**DOI:** 10.3390/s22134656

**Published:** 2022-06-21

**Authors:** Jianqu Chen, Xunmeng Li, Kai Wang, Shouyu Zhang, Jun Li, Jian Zhang, Weicheng Gao

**Affiliations:** 1College of Marine Ecology and Environment, Shanghai Ocean University, Shanghai 201306, China; m190400972@st.shou.edu.cn (J.C.); xmli@shou.edu.cn (X.L.); syzhang@shou.edu.cn (S.Z.); m180400849@st.shou.edu.cn (J.Z.); m190400973@st.shou.edu.cn (W.G.); 2Engineering Technology Research Center of Marine Ranching, Shanghai Ocean University, Shanghai 201306, China; 3Key Laboratory of Marine Ecological Monitoring and Restoration Technologies, MNR, East China Sea Environmental Monitoring Center, Shanghai 201206, China; lij@ecs.mnr.gov.cn

**Keywords:** intertidal zone, dominant species, spectral analysis, machine learning, spectral classification

## Abstract

Probing the coverage and biomass of seaweed is necessary for achieving the sustainable utilization of nearshore seaweed resources. Remote sensing can realize dynamic monitoring on a large scale and the spectral characteristics of objects are the basis of remote sensing applications. In this paper, we measured the spectral data of six dominant seaweed species in different dry and wet conditions from the intertidal zone of Gouqi Island: *Ulva pertusa*, *Sargassum thunbergii*, *Chondrus ocellatus*, *Chondria crassiaulis* Harv., *Grateloupia filicina* C. Ag., and *Sargassum fusifarme*. The different seaweed spectra were identified and analyzed using a combination of one-way analysis of variance (ANOVA), support vector machines (SVM), and a fusion model comprising extreme gradient boosting (XGBoost) and SVM. In total, 14 common spectral variables were used as input variables, and the input variables were filtered by one-way ANOVA. The samples were divided into a training set (266 samples) and a test set (116 samples) at a ratio of 3:1 for input into the SVM and fusion model. The results showed that when the input variables were the normalized difference vegetation index (NDVI), ratio vegetation index (RVI), *V*_re_, *A*_be_, *R*_g_, *L*_re_, *L*_g_, and *L*_r_ and the model parameters were g = 1.30 and c = 2.85, the maximum discrimination rate of the six different wet and dry states of seaweed was 74.99%, and the highest accuracy was 93.94% when distinguishing between the different seaweed phyla (g = 6.85 and c = 2.55). The classification of the fusion model also shows similar results: The overall accuracy is 73.98%, and the mean score of the different seaweed phyla is 97.211%. In this study, the spectral data of intertidal seaweed with different dry and wet states were classified to provide technical support for the monitoring of coastal zones via remote sensing and seaweed resource statistics.

## 1. Introduction

Seaweeds are widely distributed along 25% of the world’s intertidal rocky coastlines [1] and can absorb dissolved carbon dioxide in seawater through photosynthesis and release oxygen at the same time [2]. Their growth rates and annual primary production are much higher than those of most terrestrial plants [3]. Seaweed has a strong carbon sequestration capacity, promotes the collection of atmospheric dioxide in seawater, and is one of the important contributors of “blue carbon” [4,5]. China’s coastal intertidal zone is rich in seaweed resources, mainly *Ulva pertusa*, *Sargassum thunbergii*, *Chondrus ocellatus*, and *Sargassum fusifarme*. They mainly belong to Chlorophyta, Ochrophyta, and Rhodophyta [6]. In recent years, due to the dual influence of man-made factors, such as near-shore sewage discharge and rough seaweed harvesting, and natural factors, such as global warming and ocean acidification, the amount of seaweed resources in China’s coastal zone has been decreasing sharply year by year [7].

The Intergovernmental Panel on Climate Change (IPCC) has issued a landmark report showing temperature changes within the range of 1.5 °C. Seaweed is an important nearshore source of blue carbon, and more researchers have started to pay attention to seaweed estimates [8,9]. Additionally, research has carried out the thematic mapping of seaweed in intertidal and subtidal zones using satellite and unmanned aerial vehicle (UAV) remote sensing [10,11,12]. Spectral analysis, a basic marine remote sensing technique, is a prerequisite for remote sensing feature identification [13]. Hyperspectral data provide continuous band reflectance, making it easy to distinguish between different feature characteristics [14,15,16]. For example, some researchers conducted a comparative analysis of the spectral characteristics of nearshore corals and the domain seaweed at Lee Stockton, Bahamas, in the Caribbean, and explored the spectral variability between the dominant corals and seaweed, providing a method for coral identification using remote sensing technology [17,18]. Until now, hyperspectral techniques have been widely used in the study of terrestrial plants but have rarely been seen in the study of seaweed [19,20,21]. Zheng [22] measured the spectral data of seawater, *Ulva prolifera* and *Sargassum* in the Yellow and East China Sea and derived the spectral characteristics of the three and calculated the area covered by *Ulva prolifera*. In addition, many spectroscopy studies have been conducted on seaweed, but they are not precise to specific seaweed “species” [10,23,24,25]. On the other hand, since the study subjects are distributed in the intertidal zone, they vary over time, showing different wet and dry states, something that makes them different from exclusively terrestrial or aquatic plants. To provide theoretical guidance for the assessment of intertidal seaweed resources, this study was conducted by classifying seaweeds in different wet and dry states (with the dry-out time of two high tides). Previous studies have carried out spectral analysis of seaweed types by differential and principal component analysis (PCA) and have obtained the spectral curves and classification standards of different seaweed varieties. In this study, seaweed is classified along a spectrum, and previous methods are used for sampling and preprocessing and are combined with machine learning methods (SVM and fusion model) for automatic seaweed spectrum classification, and the parameters of the model are optimized. In addition, combined with one-way ANOVA, the input variables are screened to improve the operation efficiency of the model.

The intertidal seaweed resources along the coast of Gouqi Island in Shengsi County, Zhejiang Province, China, are relatively abundant, and the dominant seaweed types are *Sargassum thunbergii*, *Ulva pertusa*, *Chondrus ocellatus*, and *Sargassum fusifarme* [26]. To investigate the characteristics of the intertidal seaweed community of Gouqi Island, this paper investigates the spectral characteristics of six dominant seaweed species (*Sargassum thunbergii*, *Ulva pertusa*, *Chondrus ocellatus*, *Chondria crassiaulis*, *Sargassum fusifarme*, and *Grateloupia filicina*) from different phyla and at different wet and dry states. The spectral reflectance characteristics of the different seaweeds were analyzed, and 14 common spectral variables were screened by variance analysis combined with variable reduction. The screened spectral variables were used to classify the seaweeds, and a discrimination model was established for the identification of different seaweed species to provide technical support for the remote sensing monitoring of intertidal seaweeds. To fit the actual situation, the spectra of six seaweeds at different dry and wet states were also considered. Three dry states were considered: measured immediately after collection (wettest), until two high-tide periods had passed after collection (driest) and measured at any time within the waiting time between two high-tide periods (moderately dry and wet).

## 2. Materials and Methods

### 2.1. Sample Collection

Located in the south of the Ma’an Islands in Shengsi County, Zhejiang Province, China, Gouqi Island has many reefs and rich biological resources in the nearby waters as well as a unique natural geomorphology and intertidal biodiversity features. The nearshore substrate is mostly rocky reefs, where many types of seaweed grow, forming a complex nearshore marine ecosystem [27]. The seaweeds of Gouqi Island are abundant in the summer and autumn and decline in the winter and spring every year. The survey period of this study was between 17–24 October 2019, and 1–7 January 2021, from 11:30 to 13:30 (UTC/GMT+08:00) on sunny days. Human activities, such as domestic sewage discharge and mussel farms, absorb nutrients from seawater and affect ocean currents and also have an impact on seaweed growth [28,29]. Miaogan Village (122.793125 E, 30.723037 N) and Houtou Bay (122.777962 E, 30.727135 N) were selected as the seaweed spectral data collection sites (Figure 1).

Spectral data collection of the dominant intertidal seaweed species was performed using an ASD Field Spec Handheld (Field Spec Handheld, Analytical Spectral Devices (ASD), inc., Boulder, CO, USA). It has a wavelength observation range of 325–1075 nm and can observe both the visible and near infrared bands that are widely used in vegetation research and has a spectral sampling interval of 1nm, a spectral resolution of 3 nm, and a field of view of 25°.

Spectrometer optimization was performed every 10–15 min, and dark currents were collected every 5 min [30,31]. Before carrying out the spectral measurements of the features, the spectrometer needs to be calibrated against a reference whiteboard to obtain a horizontal straight line with a reflectivity of 1, and then the spectral measurement of the target feature can be performed. After the successful completion of spectrometer optimization, dark current acquisition, and whiteboard correction, the spectrometer can be pointed at the target feature, and the spectral data of the target feature can be collected and stored in real time. During spectral determination, three measurement points were taken by measuring the lower, middle, and upper sections of the thallus, and five sets of spectral data were read at 10s intervals for each measurement point. These measurements were averaged to represent the samples. A total of 400 target spectral data were collected in this study. After completing the in situ spectral measurements, samples of the experimental seaweeds were collected, brought back to the laboratory to determine their biological parameters (weight, length, dry or wet state of seaweed, etc.), and recorded.

### 2.2. Sample Preparation

One-way ANOVA was used to determine if there were statistically significant differences among each of the spectral variables between the seaweeds (Appendix A, Table A1). The 14 commonly used spectral variables were screened by one-way ANOVA. The original data were analyzed according to their variance, and the significance between different seaweed species (P, Appendix A, Table A2) was obtained. The smaller the P, the greater the significance. The 14 variables (Table 1) were used as input variables to derive the initial discriminant accuracy. The corresponding variables were eliminated from large to small according to the P until the model reached its optimal discriminative accuracy.

### 2.3. Statistical Analysis

Spectral data measured in the UV band before the 400 nm wavelength are noisy, and those measured after 900 nm are affected by water vapor absorption and should be eliminated [31]. The spectral curve of the seaweed measured every 10 times was averaged as the accurate spectral curve to reduce the influence of noise and randomness.

The spectral reflectance curves of the seaweed (six total seaweed species) were represented by PCA using the first principal component, which was used to analyze the similarities and differences among the seaweed spectra. In order to study the changes in the 14 spectral variables of the different seaweeds, the first principal component of each spectral variable in the different seaweed species was calculated, and the corresponding characteristics of each spectral variable of the different seaweeds were analyzed using this value. The data were analyzed by SPSS version 25 for Windows (SPSS Inc., Chicago, IL, USA). The comparison of the different groups was performed by one-way ANOVA, followed by Tukey’s post hoc test for multiple comparisons. The spectral variables were screened by the P number and used for classification.

Machine learning methods can directly apply the raw spectral data from the features to modeling and prediction applications and use the overall characteristics of the raw spectra as the discriminative basis for substrate feature classification. SVM is a supervised learning method for the binary classification of data, and its decision boundary is the maximum margin hyperplane solved for the test set, which allows the dimensionality of high-dimensional data to be reduced [32]. It also has the advantages of a small sample size, generality, and robustness [33]. Zhang et al. [34] used different classification methods, such as a spatially adaptive full variance method based on multiple logistic regression and a spatial feature extraction method based on super pixels, to classify spectral remote sensing images and concluded that SVM algorithms using only spectral information can effectively differentiate spectral data. The method of selecting variables can effectively reduce the amount of computation required. In this paper, we apply the soft-margin algorithm to build a SVM to distinguish seaweed spectral data quickly and accurately [35]. In order to further illustrate the reliability of the results, XGBoost was used to establish a fusion model with SVM in vote mode to classify the optimal variables. To evaluate the multi-classification classifier, the accuracy, micro-average, and macro-average receiver operating characteristic curve (ROC curve) are used to evaluate the classification effect of the model.
(1)$$\mathrm{Accuracy}=\frac{\mathrm{TP}+T}{\mathrm{TP}+\mathrm{FP}+\mathrm{TN}+\mathrm{FN}}$$
(2)$$\mathrm{Micro}-\mathrm{average}=M\left(y,\hat{y}\right)$$
(3)$$\mathrm{Macro}-\mathrm{average}=\frac{1}{\left|L\right|}\underset{l\in L}{\sum }M\left(y_{1},\hat{y_{1}}\right)$$

TP represents positive examples that have been correctly labeled as positive, and FP represents negative examples that have been incorrectly labeled as positive. TN represents negative examples that have been correctly labeled as negative, and FN represents positive examples that have been incorrectly labeled as negative. M is the evaluation measure [36]. L is the set of labels. l is the subset of L. y is the predicted set. $\hat{y}$ is the true set. $\hat{y_{1}}$ is the subset of y with label l. yl is the subset of y with label l [37].

## 3. Results

The original spectral features of the seaweed types were analyzed separately and, based on this, more extensive data were applied for separability analysis and to establish discriminatory criteria (Figure 2 and Figure 3).

### 3.1. Spectral Characteristics of Six Species

The spectral curves of the six seaweeds above were measured using an ASD spectrometer. After preprocessing the spectral data of the seaweeds in different dry and wet states, the spectral curves of the different seaweeds were drawn (Figure 3). The different lines in the figure represent the average value of a certain seaweed after 10 measurements, as shown in (a), in which each line represents the spectral values of the different dry and wet states of *Ulva pertusa*.

Six species had low reflectance in the visible wavelength band; at the wavelength of 554 nm, *Ulva pertusa* shows a reflection peak; at the wavelengths of 596 nm and 643 nm, *Sargassum thunbergii* and *Sargassum fusifarme* show reflection peaks; at 648 nm and 678 nm, *Chondrus ocellatus* showed a reflection peak, and the reflectance of those peaks was within 30% of one another. In the near-infrared band, the reflectance of the six seaweed species increases suddenly, up to 80%, and the reflectance is higher in the infrared band.

In the range of 400–700 nm, the spectral reflectance curve of *Ulva pertusa* has a peak. The reflectance of the blue-violet band (400–492 nm) is the lowest, only 3.6–7.72%; in the yellow-green band (492–597 nm), due to the reflection of the chlorophyll in the seaweed, the spectral curve has a broad and prominent high value, about 25.27%; in the orange-red band (597–700 nm), the reflection first decreases and then increases, with a minimum value of 5.78 at 669nm; in the red-edged band (670–760 nm), there is a sharp increase in the reflectivity to 83.4% (Figure 3).

In the range of 400–700 nm, *Sargassum thunbergii* and *Sargassum fusifarme* demonstrate three maximum reflectance values. In the blue-violet band (400–492 nm), the reflectivity is low, about 0.85–1.4%; in the yellow-green band (492–597 nm), the reflectivity increases continuously and reaches its maximum at 596 nm, which is about 3.74%; in the orange-red band (597–700 nm), there is another maximum at 643 nm, and the reflectivity is about 2.19%. In the red-edged band (670–760 nm), the reflectivity increases sharply, and is about 34.91% at its highest. The reflectivity in the short wave band (780–900 nm) is about 35%.

In the 400–700 nm range, the total reflectance of *Chondrus ocellatus*, *Chondria crassiaulis*, and *Grateloupia filicina* is high in the visible light range, and there are two maxima. In the blue-violet band (400–492 nm), the reflectivity is about 30.4–33.79%; in the yellow-green band (492–597 nm), the reflectance first decreases and then increases, and there is a minimum at 536 nm, which is about 12.55%; in the orange-red band (597–780 nm), there are two maxima at 648 nm and 678 nm, and the reflectivity is about 33.87% and 32.31%, respectively. In the red-edged band (670–760 nm), the reflectivity rises sharply and is about 86.04% at its maximum. In the near-infrared short wave band (780–900 nm), the reflectivity is stable at about 85%.

### 3.2. Trend Analysis of Seaweed Spectra

Each seaweed was subjected to PCA, and their loadings were all greater than 90%, indicating that the first principal component can effectively express the information in each seaweed dataset. The positions of the green peak, red valley, blue edge, and red edge are marked on the spectral curves, as shown in Figure 4. To observe the trends in the spectral variables more clearly, the four regions corresponding to the reflection spectra and the first-order derivative spectra in regions (a), (b), (c), and (d) are enlarged in Figure 5 to obtain four spectral enlargements.

It can be seen from Figure 3 and Figure 4 that there are obvious differences in the seaweed spectra among the different phyla, while there are small differences among the same species (PCA loadings greater than 90%). For example, in (b), the green peak amplitudes *R*_g_ in Rhodophyta, *Chondria crassiaulis*, *Chondrus ocellatus*, and *Grateloupia filicina* were −0.8063, −0.8247 and −0.8158, respectively; in Ochrophyta, the values of *Sargassum thunbergii* and *Sargassum fusifarme* were −0.7590 and −0.7555, respectively. The value of *Ulva pertusa* was −0.5706. The locations of the green peak location, *L*_g_: in Rhodophyta, they were 535.6500 nm, 521.7119 nm, and 526.5200 nm for *Chondria crassiaulis*, *Chondrus ocellatus*, and *Grateloupia filicina*, respectively; in Ochrophyta, *Sargassum thunbergii* and *Sargassum fusifarme* had values of 559.8025 and 560.0000, respectively; The value of *Ulva pertusa* was 551.1475. The difference of spectral curves among the same phylum are smaller than those among different phyla.

### 3.3. Analysis and Optimal Screening of Spectral Variables of Seaweeds

The results of the one-way ANOVA (Appendix A, Table A2) were averaged to represent the P of the corresponding variables (Table 2). The 14 variables were selected according to significance, and the variables demonstrating low levels of significant difference were eliminated one by one until the model reached its optimal discriminant accuracy; the order of significance of the 14 variables was as follows: NDVI (*R*_g_, *R*_r_), RVI (*R*_g_, *R*_r_), *V*_re_, *L*_be_, *A*_be_, *L*_g_, *L*_r_, *R*_r_, *L*_re_, *R*_g_, NDVI (*A*_re_, *A*_be_), *A*_re_, *V*_be_, RVI (*A*_re_, *A*_be_) (Table 2).

### 3.4. Support Vector Machine Classification

Because the growth environment of intertidal seaweeds is different from that of terrestrial and aquatic plants, the rise and fall of the tide affects the spectral reflection curves of seaweed. In order to eliminate the influence of seawater on the discrimination results and to expand the application scenarios, it was necessary to carry out spectral determination tests of seaweeds at different degrees of dryness. Among them, class 1 represents *Chondrus ocellatus*, class 2 represents *Chondria crassiaulis*, class 3 represents *Grateloupia filicina*, class 4 represents *Sargassum thunbergii*, class 5 represents *Sargassum fusifarme*, and class 6 represents *Ulva pertusa*. A total of 382 groups of data were randomly divided into a training set and a testing set at a ratio of 3:1.

In this paper, the construction of the SVM-based classification model was based on PyCharm in the Python 3.7 environment and mainly used the joblib module of the scikit-learn SVM (sklearn SVM). The radial basis function (RBF) was used as the kernel function, and its value was set to 2; degree = 3; coef0 = 0.

As can be seen from Table 3, when all 14 variables were used to distinguish between the seaweed species, the effect was the worst, only 40.89%; when the variables with smaller significance (larger P) were eliminated one by one, the discrimination accuracy increased. Until the four variables with the lowest significance were eliminated, the highest discrimination accuracy was 74.99%. Then, as the number of input variables decreased, the discrimination accuracy also decreased. In order to explore the best model parameters, the parameters of each model were optimized. Using the grid-search method (CV = 4), the optimal input parameters of the SVM model were determined, and the relationship between the parameters and accuracy was obtained (interspecific, Figure 6).

When the NDVI (*R*_g_, *R*_r_); RVI (*R*_g_, *R*_r_); *V*_re_, *A*_be_, *R*_g_, *L*_re_, *L*_g_, and *L*_r_ were selected as the input variables, the optimal variables of the model were g = 1.30 and c = 2.55, and the accuracy was 74.99%. By removing the misclassified seaweed species, it was found that there was more misclassification between Rhodophyta and Ochrophyta, especially between different dry and wet conditions. No seaweed classification errors were observed among the different phyla. Using the grid-search method (CV = 4), the optimal input parameters of the SVM model were found and relationship between the parameters and accuracy was obtained (between phyla, Figure 7).

When the same input variables were selected and the model parameters of different phyla (Chlorophyta: *Ulva pertusa*; Ochrophyta: *Sargassum thunbergii*, *Sargassum fusifarme*; Rhodophyta: *Chondrus ocellatus*, *Chondria crassiaulis*, *Grateloupia filicina*) were optimized, the results showed that when g = 6.85 and C = 2.55, the discrimination accuracy was the highest, achieving a value of 93.94%.

Combined with the above analysis, the best SVM model input variables and model parameters were applied and used for classification. The corresponding classification confusion matrix is shown in Figure 8a, and the ROC curves of the phyla and interspecific classification are shown in Figure 8b,c.

The evaluation indicators calculated by the macro-average method considered all of the classes to be of equal importance. Under each category, the probability of the m test samples can be obtained for each category. Therefore, according to each corresponding column in the probability matrix and the label matrix, the false positive rate (FPR) and the true positive rate (TPR) under each threshold can be calculated to create an ROC curve. In this way, the total of n the ROC curves can be drawn. Finally, the final ROC curve can be obtained by averaging the n ROC curves (macro-averaged ROC). From Figure 8b,c, the area under the curve (AUC) corresponding to the macro-averaged ROC curve of the phyla and seaweed species is 0.87 and 0.90, respectively. The micro-average-calculated evaluation indicators consider the contribution of each sample. The AUC corresponding to micro-averaged ROC curve of the phyla and seaweed species is 0.89 and 0.92, respectively. The AUC value corresponding to each ROC curve is greater than 0.75. The AUC of *Ulva pertusa* and Chlorophyta is 1.00, as shown in Figure 8a (the predicted label of Category 2 in the confusion matrix is all equal to the actual label). Rhodophyta has the lowest AUC. *Chondria crassiaulis*, *Chondrus ocellatus*, *and Grateloupia filicina* belong to classes 0, 1, and 4, and have AUC values of 0.75, 0.84, and 0.99, respectively.

### 3.5. Fusion Model Classification

Because the classification results of only a single model are unsatisfactory, we used the model fusion method to further verify the six seaweed varieties. SVM and XGBoost were fused by the vote algorithm to obtain the fusion model. By bringing the variables and parameters filtered by SVM into the fusion model, all of samples were divided into a training set and a test set at a ratio of 3:1, and the classification results and accuracy were calculated. The vote score mean was 97.211%, and the classification results are shown in Figure 9.

From the classification results, the classification of Category 1 (*Ulva pertusa*) is correct. There are 20 mismatches in Category 2 (*Sargassum thunbergii*) and in Category 3 (*Sargassum fusifarme*). There are 12 mismatches in Category 4 (*Chondrus ocellatus*), category 5 (*Grateloupia filicina* C. Ag.), and category 6 (*Chondria crassiaulis* Harv.). The overall accuracy is 73.98%, which further confirms the difficulty of spectral data in processing when classifying members of the same seaweed species.

Similarly, the best SVM model input variables and model parameters were applied and used for classification, and the corresponding classification confusion matrix is shown in Figure 10a. The ROC curves of the phyla and interspecific classification are shown in Figure 10b,c.

The AUC corresponding to the macro-averaged ROC curve of the phyla and seaweed species is 0.88 and 0.90, and the micro-averaged ROC is 0.89 and 0.91, respectively. The AUC of *Ulva pertusa* and Chlorophyta is 1.00. Rhodophyta has the lowest AUC. *Chondria crassiaulis*, *Chondrus ocellatus*, and *Grateloupia filicina* are 0.80, 0.84, and 0.99, respectively.

## 4. Discussion

In this study, after the spectral characteristics of each seaweed variety had been analyzed, they were processed by differential transformation, and the characteristic variables were selected according to the spectral characteristics that were stable in a certain range. The green peak of *Ulva pertusa* appeared near 528 nm, and the results were consistent with the spectral characteristics of green vegetation [38]. Chlorophyll absorbs blue light and red light and does not absorb green light, but the reflectivity increases when the wavelength is greater than 700 nm. The reflection mechanism of each cell is similar to a small corner reflector, so the cell’s structure is also an important factor that affects reflectivity [39]. Therefore, the electromagnetic reflectance of *Ulva pertusa* in the red-edged band can rapidly increase from 5% to 80%. Due to the lack of active fluorescence absorption on the surface of seaweed [40], the reflectance of the near infrared short-wave band (780–900 nm) is about 80%.

The reflectivity of *Sargassum thunbergii* and *Sargassum fusiforme* is low in the visible range, and there are three maxima. There are maxima at 570 nm, 596 nm, and 643 nm, and the results are consistent with those of drifting seaweed (*Sargassum*) [22]. It shows the applicability of this method and the reliability of the spectrum of vegetation in this study. Although there are few studies on the spectral characteristics of Rhodophyta, other studies (such as those on corals) have found Rhodophyta on the surface of bleached coral [40,41]. Through comparative analysis, it is not difficult to determine that the spectral characteristics of Rhodophyta are very similar.

According to Table 1-1 in Graham and Wilcox’s textbook, *Algae* (2000), the elements of different phylum are different. The spectral waveforms of each seaweed are similar to those of the corresponding phylum that have been reported in many studies. Generally, the distribution of seaweed offshore is as follows: the distribution of green seaweed is the shallowest, followed by brown seaweed, and red seaweed is often in the deepest water [41]. Seaweed growing in shallow coastal zones has evolved to have specific mechanisms to resist damage from strong light. The spectral curves of *Sargassum thunbergii* and *Sargassum fusifarme* are very similar to those of the reported seaweed varieties. There are fine bimodal patterns at 600 nm and 650 nm, and the maximum reflectivity in the near-infrared and infrared bands is 40% [10,22,24,42] (Figure 3).

As the thickness of *Ulva pertusa* increases, the reflectance of *Ulva pertusa* and *Ulva prolifera* is similar, and the reflectance of the yellow-green band and near-infrared band increases as the seaweed thickness increases [22,43]. In this paper, the multi-layered spectral data of *Ulva pertusa* were measured. The results showed that as the number of layers increased, the spectral curve of *Ulva pertusa* was about 550 nm, and the near-infrared band and the reflectance increased exponentially. When it was superposed to three layers, the spectral reflectance of the yellow-green band was the highest, about 25%, and the near-infrared band was about 80%. When the spectral reflectance curve of *Ulva pertusa* increased to includes four, five, and six layers, it was similar to the curve obtained with three layers, indicating that the spectral reflectance curve of *Ulva pertusa* reached saturation state when at three layers, which was very similar to that of *Ulva prolifera* but with a different thickness. Liu Qing studied the physiological and biochemical responses of intertidal seaweed to copper stress, and the results showed that the chlorophyll content decreased as Cu^2+^ content increased in *Ulva pertusa* [44]. Therefore, the chlorophyll content in *Ulva pertusa* (yellow-green band) can be used as an indicator of heavy metal pollution in coastal zones.

To sum up, the chlorophyll and carotenoid contents are similar in the same phylum, and the differences among different phylum are obvious. Therefore, it should be considered that the same spectrum phenomenon may occur for foreign matter among the same phylum as well. The higher the plant activity, the better the chlorophyll activity and the higher the spectral reflectance in the corresponding band [45], so it is necessary to measure the spectrum of the same seaweed in different seasons to further explore the seasonal variations in the spectral reflectance curves of seaweed. In addition, it has been confirmed that light adaptation is related to sex differences in seaweed [46]. However, the effect of seaweed sex on the spectrum is not considered in this paper, and a follow-up study is needed to supplement this information.

In this paper, spectral variables with lower P values were eliminated one by one, and the classification accuracy was gradually improved. When the NDVI (*R*_g_, *R*_r_), RVI (*R*_g_, *R*_r_), *V*_re_, *A*_be_, *R*_g_, *L*_re_, *L*_g_, and *L*_r_ were used as input variables, the accuracy of the model was the highest. A support vector machine with a soft margin algorithm was used to classify the seaweed spectra. At the same time, XGBoost was mixed with SVM in vote mode to classify the same datasets. The input variables of this spectral curve were obtained by the same processing method as the ones used for the other seaweed varieties. The distribution range of these two seaweed varieties is small. Due to their small volume, they are neither common nor dominant species, so it is difficult to collect and analyze them, and they were not collected in the experimental stage. In order to fill the gap found in green seaweed, we obtained the spectral curves of *Ulva prolifra* by searching the literature [22]. Additionally, as it belongs to Category 7, it was substituted into the classification model for operation. Because there was only one sample, the training set could not be set, but misclassifications were only observed in *Ulva pertusa* (Chlorophyta), and not in Rhodophyta or Phaeophyta. As shown in Figure 9, the fault diversity of red seaweed falls in Rhodophyta, and the fault diversity of brown seaweed falls in Phaeophyta. The results are basically consistent: the classification effect of seaweeds from the same phylum was better (93%), and the classification effect of seaweeds from different species was worse (74%).

The spectral feature analysis, response, and screening of characteristic variables and the classification model established here are based on ground hyperspectral data, but they can be combined with near-ground low-altitude remote sensing, aerospace remote sensing and aerospace remote sensing, or satellite remote sensing to correct their spectra and establish a spectrum database after score analysis by using spectral features to construct relevant variables for remote sensing inversion, identification, and monitoring [47,48,49]. This study only includes the hyperspectral data of the seaweed collected in the intertidal zone of Gouqi Island, Zhejiang Province, in autumn and winter 2019–2020. Different growth environments may lead to different species having different hyperspectral response characteristics, so it is necessary to further study the discrimination models in different regions.

## 5. Conclusions

In this paper, the spectral data of *Sargassum fusifarme*, *Ulva pertusa*, *Chondria crassiaulis*, *Sargassum thunbergii*, and *Grateloupia filicina* collected from Gouqi Island were obtained by principal component analysis, which can provide base data for the remote sensing monitoring of fisheries. Through the first-order differential analysis, the differences of the spectral characteristics of six seaweed varieties were obtained. The spectral reflectance curves of the seaweeds from the same phylum were very similar, which may be caused by differences in the contents of different elements. In addition, the results of the one-way ANOVA showed that the differences between seaweeds of the same phylum were not significant (*p* > 0.05), while the differences between seaweeds from different phyla were significant (*p* ≤ 0.05).

A SVM classification model was used in collaboration with the manual identification method to identify different seaweed species and to improve the identification efficiency and accuracy. When the NDVI (*R*_g_, *R*_r_), RVI (*R*_g_, *R*_r_), *V*_re_, *A*_be_, *R*_g_, *L*_re_, *L*_g_, and *L*_r_ were selected as input variables, the SVM model constructed with the Gaussian kernel function was better able to distinguish between the six seaweed species with an accuracy of 74.96% when the model parameters were c = 1.30 and g = 2.85. Through the fusion model comprising XGBoost and SVM, the variable screening method was used, and when the same optimal variables were used with SVM, the accuracy was the highest: 73.98%. Additionally, many types of errors were observed in seaweeds within the same phylum. When there were SVM misclassifications, the differences in the seaweed spectrum among varieties from the same phylum was small, and the level of misclassification was large; however, the difference in the seaweed spectrum among varieties from different phyla was large, and the error rate of misclassification was lower. The results are similar to those of the fusion model. Using the same method to distinguish the seaweed phyla (Chlorophyta, Ochrophyta, and Rhodophyta), an accuracy of 93.94% was achieved.

In this paper, dominant seaweed species with different degrees of dryness and wetness from Gouqi Island underwent spectral analysis and classification, which have good practicability, in order to provide technical support and provide a partial database for the remote sensing of intertidal seaweed. The element and spectral response mechanisms of different species from the same phylum need to be studied further.

## Figures and Tables

**Figure 1 sensors-22-04656-f001:**
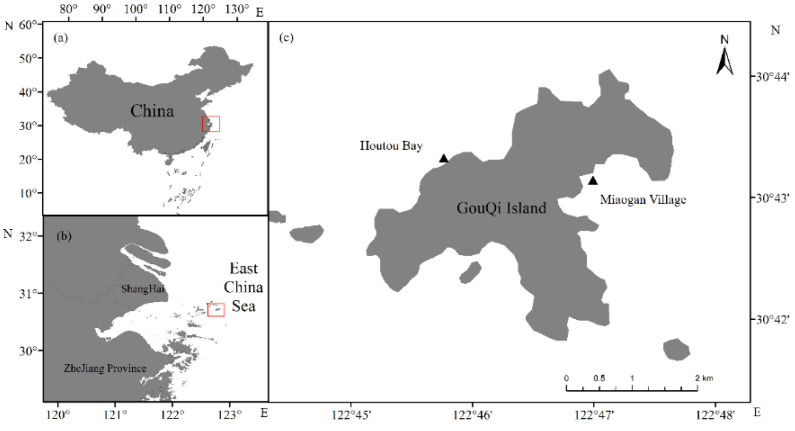
Map of the research area: (**a**) China; (**b**) Yangtze River estuary; (**c**) survey sites. Triangles indicate sampling locations.

**Figure 2 sensors-22-04656-f002:**
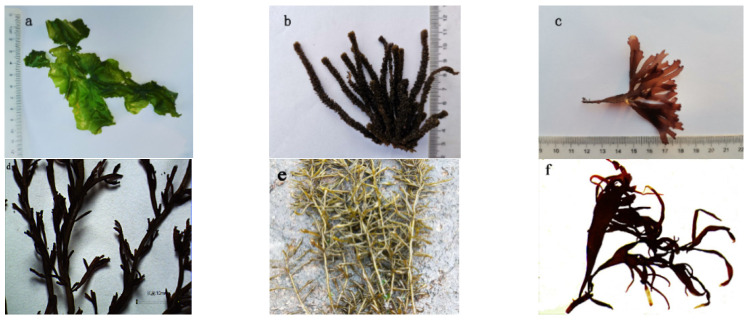
Dominant species in the intertidal zone of Gouqi Island: (**a**) *Ulva pertusa*; (**b**) *Sargassum thunbergii*; (**c**) *Chondrus ocellatus*; (**d**) *Chondria crassiaulis*; (**e**) *Sargassum fusifarme*; (**f**) *Grateloupia filicina*.

**Figure 3 sensors-22-04656-f003:**
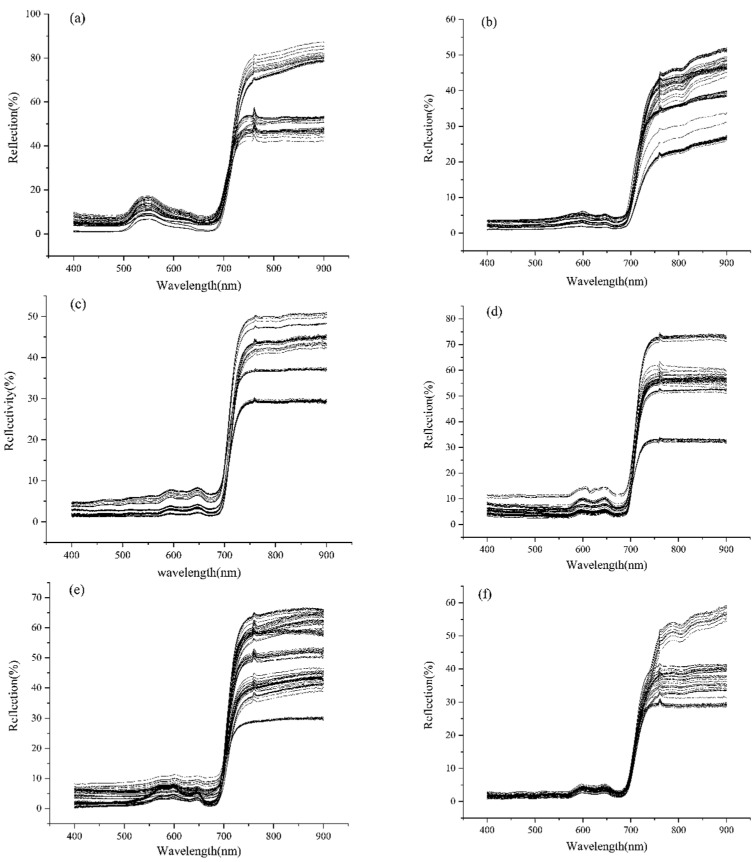
Spectral curves of six species of seaweed: (**a**) *Ulva pertusa*; (**b**) *Sargassum thunbergii*; (**c**) *Chondrus ocellatus*; (**d**) *Chondria crassiaulis*; (**e**) *Sargassum fusifarme*; (**f**) *Grateloupia filicina*; In (**a**–**f**), different lines represent different seaweed samples.

**Figure 4 sensors-22-04656-f004:**
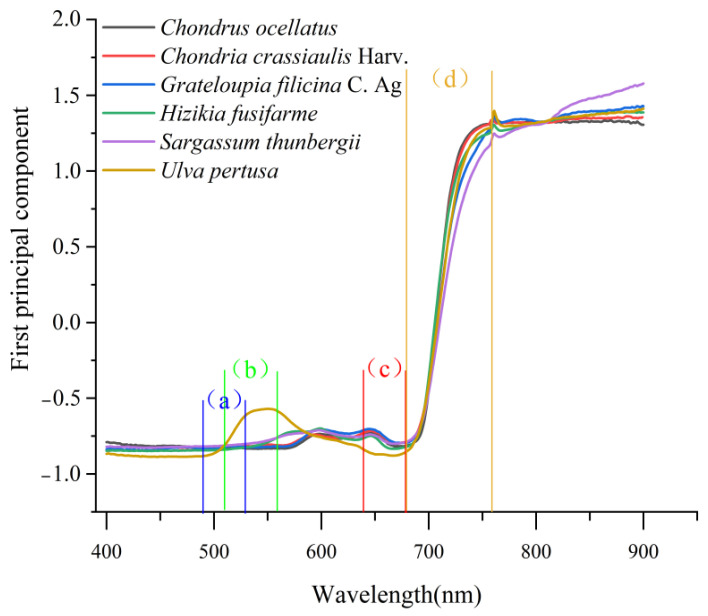
Spectral curves of the first principal component. (a) Blue border area; (b) green peak area; (c) red valley area; (d) red area. See Figure 5 for details.

**Figure 5 sensors-22-04656-f005:**
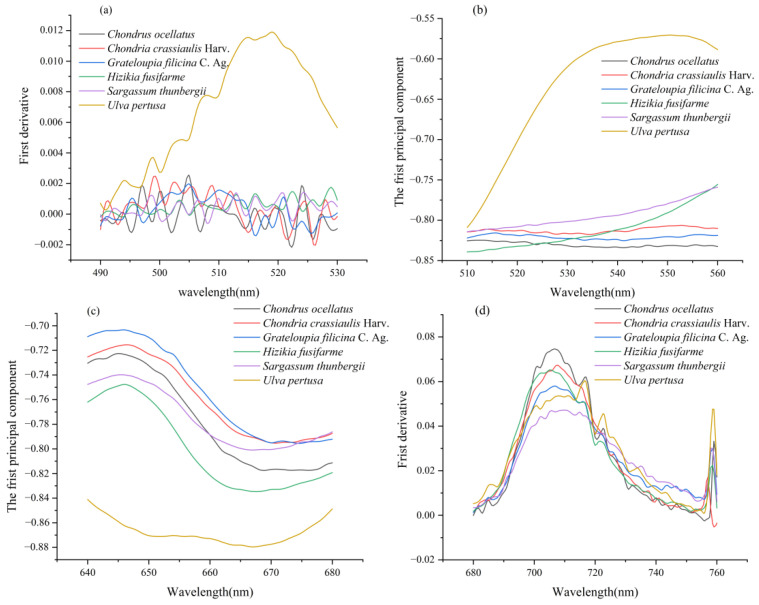
Spectrum enlarged view of green, red valley, blue edge, and red edge location. (**a**) Enlarged image of the first-order differential spectrum at 490–530 nm (blue border area); (**b**) enlarged image of spectrum at 510–560 nm (green peak area); (**c**) enlarged image of spectrum at 640–680 nm (red valley area); (**d**) enlarged image of the first-order differential spectrum at 680–760 nm (red area).

**Figure 6 sensors-22-04656-f006:**
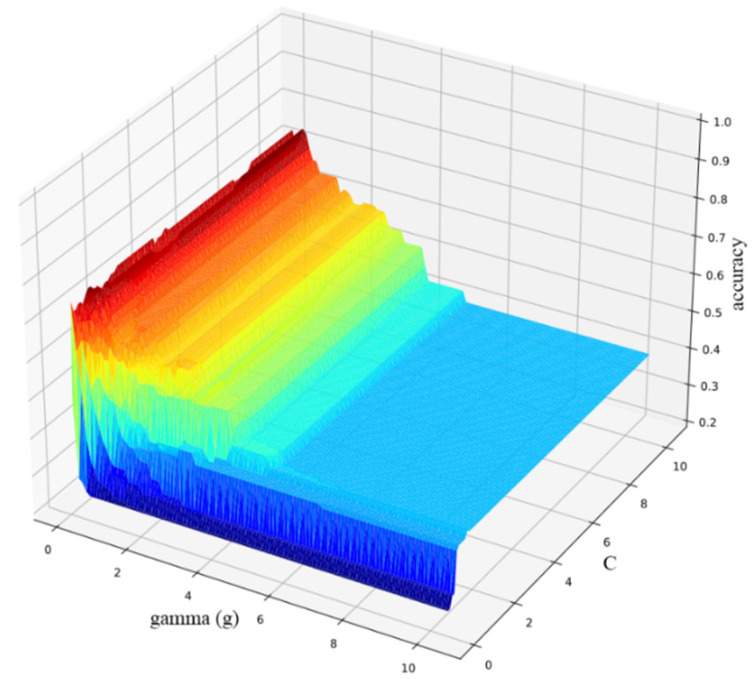
Parameter selection results (differences between seaweed species).

**Figure 7 sensors-22-04656-f007:**
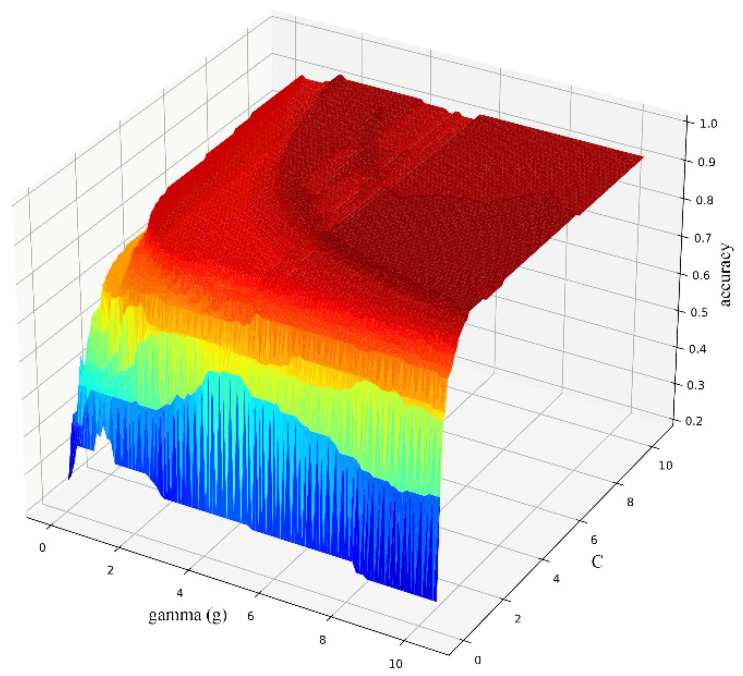
Parameter selection results (different seaweed phyla).

**Figure 8 sensors-22-04656-f008:**
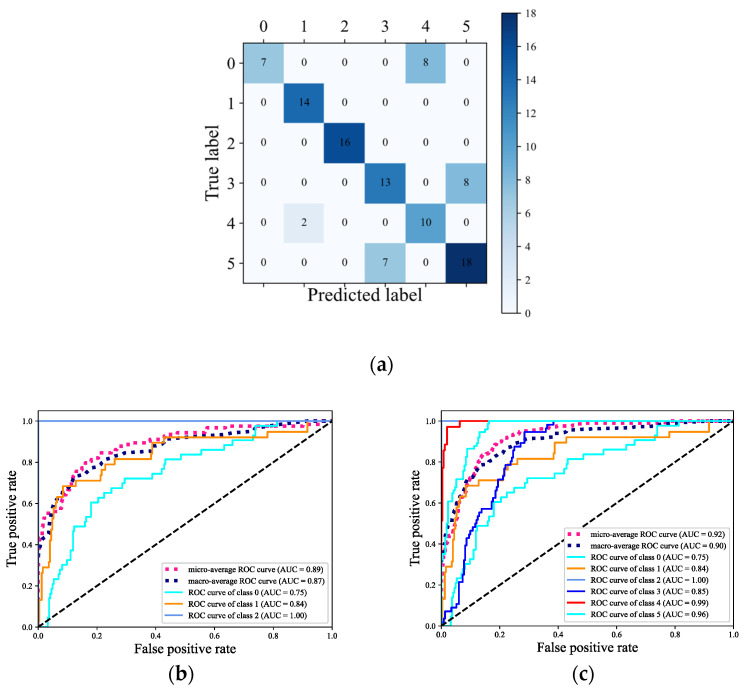
Confusion matrix and ROC curves of the optimal SVM model for the classification of six species of seaweed in the intertidal zone. 0–5 in (**a**) and classes 0–5 in (**c**) are *Chondria crassiaulis*, *Chondrus ocellatus*, *Ulva pertusa*, *Sargassum thunbergii*, *Grateloupia filicina*, and *Sargassum fusifarme*, respectively; classes 0–5 in (**b**) are Rhodophyta, Phaeophyta, and Chlorophyta.

**Figure 9 sensors-22-04656-f009:**
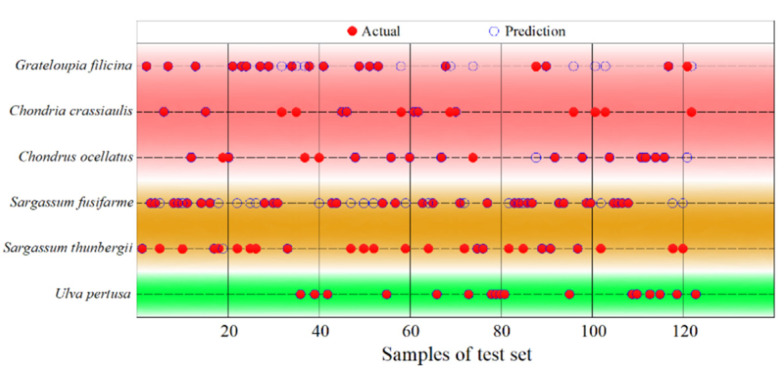
Classification results.

**Figure 10 sensors-22-04656-f010:**
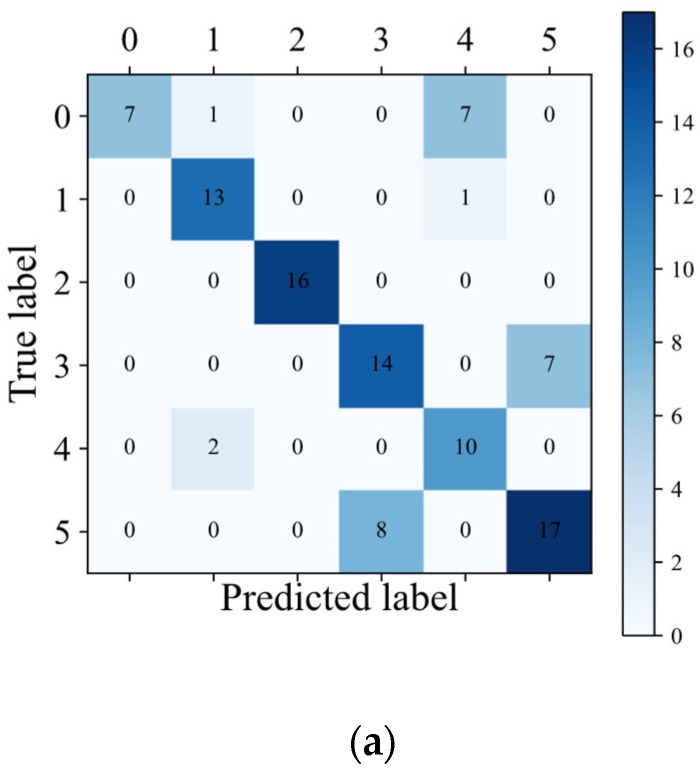
Confusion matrix and ROC curves of the optimal fusion model for the classification of six seaweed species in the intertidal zone. Species 0–5 in (**a**) and classes 0–5 in (**c**) are *Chondria crassiaulis*, *Chondrus ocellatus*, *Ulva pertusa*, *Sargassum thunbergii*, *Grateloupia filicina*, and *Sargassum fusifarme*, respectively. Classes 0–5 in (**b**) are Rhodophyta, Phaeophyta, and Chlorophyta.

**Table 1 sensors-22-04656-t001:** Fourteen commonly used spectral variables.

Variables Types	Variables	Symbol	Definition	Reference
Location variables	Green peak amplitude	*R* _g_	Maximum reflectivity of 510–560 nm in green light range	Minglu, T., 2016
	Green peak location	*L* _g_	Wavelength of green peak in the green range of 510–560 nm	Fuqi, Y., 2012
	Red valley amplitude	*R* _r_	Maximum reflectivity of 640–680 nm in red light range	Fuqi, Y., 2012
	Red valley location	*L* _r_	Wavelength corresponding to red valley at 640–680 nm in red light range	Fuqi, Y., 2012
	Red edge amplitude	*V* _re_	Maximum value of first order differential of red edge at 680–760 nm	Xiaokang, Y., 2021
	Red edge location	*L* _re_	Wavelength corresponding to red edge amplitude	Cho, M. A., 2006
	Blue edge amplitude	*V* _be_	First order differential maximum of blue edge at 490–530 nm	Xiaokang, Y., 2021
	Blue edge location	*L* _be_	Band length corresponding to blue edge amplitude	Yuna, W., 2021
Area variables	Red edge area	*A* _re_	Sum of first order differential values in the range of red edge	Xiaokang, Y., 2021
	Blue edge area	*A* _be_	Sum of first order differential values in the range of blue edge	Xiaokang, Y., 2021
Vegetation index variables	*R*_g_/*R*_r_	RVI (*R*_g_, *R*_r_)	Amplitude ratio of green peak to red valley	Minglu, T., 2016
	*A*_re_/*A*_be_	RVI (*A*_re_, *A*_be_)	Area ratio of red edge to blue edge	Xiaokang, Y., 2021
	(*R*_g_ − *R*_r_)/(*R*_g_ + *R*_r_)	NDVI (*R*_g_, *R*_r_)	Normalized ratio of green peak to red valley amplitude	Minglu, T., 2016
	(*A*_re_ − *A*_be_)/(*A*_re_ + *A*_be_)	NDVI (*A*_re_, *A*_be_)	Normalized ratio of red edge area to blue edge area	Xiaokang, Y., 2021

**Table 2 sensors-22-04656-t002:** P of different spectral variables. “*” from Table A2.

Spectral Variables	Number of “*” in Appendix A, Table A2
NDVI (*R*_g_, *R*_r_)	14
RVI (*R*_g_, *R*_r_)	14
*V* _re_	14
*A* _be_	13
*R* _g_	13
*L* _re_	12
*L* _g_	12
*L* _r_	11
*R* _r_	11
*L* _be_	11
NDVI (*A*_re_, *A*_be_)	11
*A* _re_	10
*V* _be_	9
RVI (*A*_re_, *A*_be_)	7

**Table 3 sensors-22-04656-t003:** Selection results of spectral characteristic variables.

Rejected Variables	Variable Quantity	Accuracy (%)
----	14	40.89
RVI (*A*_re_, *A*_be_), *V*_be_, *A*_re_, NDVI (*A*_re_, *A*_be_)	10	68.34
RVI (*A*_re_, *A*_be_), *V*_be_, *A*_re_, NDVI (*A*_re_, *A*_be_), *L*_be_	9	66.02
RVI (*A*_re_, *A*_be_), *V*_be_, *A*_re_, NDVI (*A*_re_, *A*_be_), *L*_be_, *R*_r_	8	74.99
RVI (*A*_re_, *A*_be_), *V*_be_, *A*_re_, NDVI (*A*_re_, *A*_be_), *L*_be_, *R*_r_, *L*_r_	7	72.03
RVI (*A*_re_, *A*_be_), *V*_be_, *A*_re_, NDVI (*A*_re_, *A*_be_), *L*_be_, *R*_r_, *L*_r_, *L*_g_	6	64.68

## Data Availability

The raw/processed data required to reproduce these findings cannot be shared at this time as the data also forms part of an ongoing study.

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
