# Peer review of "Variable Optimization of Seaweed Spectral Response Characteristics and Species Identification in Gouqi Island"

_sensors, 2022, doi:10.3390/s22134656_

Round 1
Reviewer 1 Report
The spectral characteristics of six seaweeds were obtained and analysed with SVM. Overall, it is a pleasantly well written paper with a clear and well-organized structure. I enjoyed reading this paper. I would recommend accept after only a few minor corrections listed as follows.
Line 47:
" Seaweed as an" should be " Seaweed is an"
Line 48:
"researches have " should be "research has "
Line 127:
"statistical significant " should be " statistically significant"
Line 17, 154, 402:
Define abbreviation “SVM” as the first time when it is used in the text. Also check other abbreviations. forexample the PCA was defined twice at lines 68 and 143.
Line 177:
Define abbreviation “ASD”
Author Response
Thank you for your recognition of this paper. Your comments and suggestions are very helpful to improve the quality of our manuscript and our future work. Please see the attachment for the specific reply.
Reviewer 2 Report
- The manuscript is concerned with parameter optimization of seaweed spectral response characteristics and species identification in Gouqi Island, which is interesting. It is relevant and within the scope of the journal.
- However, the manuscript, in its present form, contains several weaknesses. Adequate revisions to the following points should be undertaken in order to justify recommendation for publication.
- Full names should be shown for all abbreviations in their first occurrence in texts. For example, ANOVA in p.1, NDVI in p.1, RVI in p.1, UAV in p.2, RBF in p.9, etc.
- For readers to quickly catch the contribution in this work, it would be better to highlight major difficulties and challenges, and your original achievements to overcome them, in a clearer way in abstract and introduction.
- p.1 - six seaweed species are adopted in this study. What are the other feasible alternatives? What are the advantages of adopting these species over others in this case? How will this affect the results? More details should be furnished.
- p.1 - a combination of one-way ANOVA, Support Vector Machine and, the fusion model of eXtreme Gradient Boosting and SVM are adopted to identify and analyze the different seaweed spectrums. What are the other feasible alternatives? What are the advantages of adopting these soft computing techniques over others in this case? How will this affect the results? More details should be furnished.
- p.2 - Gouqi Island is adopted as the case study. What are other feasible alternatives? What are the advantages of adopting this case study over others in this case? How will this affect the results? The authors should provide more details on this.
- p.3 - one-way ANOVA is adopted to determine if there are statistical significant differences for each spectral parameter between all the seaweeds. What are the advantages of adopting this approach over others in this case? How will this affect the results? The authors should provide more details on this.
- p.3 - fourteen parameters as shown in Table 1 are adopted as input variables to derive the initial discriminant accuracy. What are the advantages of adopting these parameters over others in this case? How will this affect the results? The authors should provide more details on this.
- p.4 - principal component analysis is adopted to represent the spectral reflectance curves of seaweed. What are the advantages of adopting this approach over others in this case? How will this affect the results? The authors should provide more details on this.
- p.4 - Tukey’s post-hoc test is adopted for multiple comparisons. What are the advantages of adopting this test over others in this case? How will this affect the results? The authors should provide more details on this.
- p.5 - the soft-margin algorithm is adopted to build a SVM. What are the advantages of adopting this algorithm over others in this case? How will this affect the results? The authors should provide more details on this.
- p.5 - XGBoost is adopted to establish a fusion model with SVM in vote mode. What are the advantages of adopting this soft computing technique over others in this case? How will this affect the results? The authors should provide more details on this.
- p.9 - RBF is adopted as a kernel function. What are the advantages of adopting this kernel function over others in this case? How will this affect the results? The authors should provide more details on this.
- p.9 - accuracy is adopted to assess the performance of the models. What are the other feasible alternatives? What are the advantages of adopting this evaluation metric over others in this case? How will this affect the results? More details should be furnished.
- p.10 - the grid-search method is adopted to find the optimal input parameters of the SVM model. What are the advantages of adopting this method over others in this case? How will this affect the results? More details should be furnished.
- p.12 - “…the elimination continues, the classification accuracy begins to decline. The reason may be that.…” More justification should be furnished on this issue.
- Some key model parameters are not mentioned. The rationale on the choice of the set of parameters should be explained with more details. Have the authors experimented with other sets of values? What are the sensitivities of these parameters on the results?
- The discussion section in the present form is relatively weak and should be strengthened with more details and justifications.
- Some assumptions are stated in various sections. More justifications should be provided on these assumptions. Evaluation on how they will affect the results should be made.
- Moreover, the manuscript could be substantially improved by relying and citing more on recent literature about real-life applications of soft computing techniques in different fields such as the following. Discussions about result comparison and/or incorporation of those concepts in your works are encouraged:
- Banan, A., et al., “Deep learning-based appearance features extraction for automated carp species identification,” Aquacultural Engineering 89: 102053 2020.
- Shamshirband, S., et al., “Prediction of significant wave height; comparison between nested grid numerical model, and machine learning models of artificial neural networks, extreme learning and support vector machines,” Engineering Applications of Computational Fluid Mechanics 14 (1): 805-817 2020.
- Fan, Y.J., et al., “Spatiotemporal modeling for nonlinear distributed thermal processes based on KL decomposition, MLP and LSTM network,” IEEE Access 8: 25111-25121 2020.
22. Some inconsistencies and minor errors that needed attention are:
- Replace “…divided into 3 dries out degrees…” with “…divided into 3 dried out degrees…” in line 88 of p.2
- Replace “…During spectral determination, Three measurement points…” with “…During spectral determination, three measurement points…” in lines 118-119 of p.3
- Replace “…SVM-base classification model…” with “…SVM-based classification model…” in line 262 of p.9
- Replace “…From table 3…” with “…From Table 3…” in line 267 of p.9
23. Some recommendations are made for further investigation. Why are they not performed in this study? More justifications should be furnished on this.
Author Response

(The authors gave the same response as above.)

Reviewer 3 Report
This paper has a number of problems that make it not suitable for publication in its current form. These can be roughly divided into two categories; content and presentation. Comments here will focus on content, but will be followed by some further comments on the presentation which is in need of improvement.
A major concern with the content of the manuscript actually follows from the title. From that title, the reader is led to think that a set of optimal parameters for classifying seaweeds will be identified, and then a classification will be presented. The former is really just confusing. Do the authors refer here to model parameters for the SVM, or to classification features from the reflectance data? It's just not clear. Also, the title implies that we will identify species using spectral data, but the corresponding results are not very clear. I'll come back to this point later.
The sample material in section 2.1is reasonably clear, although I don't think the authors want to refer to their sampling point selection as done "haphazardly" which means it was done carelessly. It's also not clear why the 14 'parameters' listed in section 2.2 were chosen. ACtually, these are not parameters at all, really. A better term, more consistent with standard usage in remote sensing spectral analysis, might be 'variable.' After carefully reading this section, I think these are what was referred to in the title. If so, then a different term should be chosen. It's also not at all clear why the anova was done, or what it's results are. I would suggest that the authors; 1) clarify why they used this technique, 2) give us an ANOVA table with F-value and statistics. It seems as though ANOVA was used to screen out the spectral indices in Table 1. Is this correct? If so, caan the resulting screenign be more clearly presented. I had no idea which of the 14 indices were considered the most effective. Beginning in the paragraph on line 142, a PCA is also described. Again, it's not clear why this was done. Figure 4 seems to suggest that it was used as a way to generalize all the spectra from a particular species. If so, why not just average them? This is just not clear. Also, Section 3.1 begins with a redundant statement about measurement. We already know this? This is one of several instances where methods are repeated. This makes for some confusion for the reader, who is left to try to decide if this is just a repeat of earlier information, or if something different has happened.
Section 3.2 is where the analysis of the spectral variables is described, but here again it's very difficult to follow what was done. Figure 4 shows the first PC, which looks to me like spectral curves, and makes me wonder whether the PCA was really necessary or appropriate. Please explain, in much more detail, what information it adds and why you chose to use this technique. In the paragraph beginning on line 232, the discussion suggests that PCAwas used as a way to visually discriminate between the various species. If this is correct, then it should be stated clearly and succinctly.
Section 3.3 is where the screening of the spectral features was done. Again, the procedure here is not well-explained, but from I could tell it came down to running a series of SVMmodels using different sets of inputs, until the best model was achieved. There's nothing really wrong with this, but there's really not much systematic information gained. It looks like the best model consisted of 8 parameters, and was 75% accurate, can we get some discussion on what those parameters were and why they were important? The material on fitting the SVMparameters is actually very good. I wish more users of SVMS would detail their results as well as this paper did.
In the final part of Section 3, we are told that some boosted models were used, and apparently, the results got much better. Unfortunately, we don't much information at all about the boosted fusion models, and classification results Figure (Fig 8) is cryptic and not very helpful. Since this is essentially a classification, I would strongly urge the authors to present their results in the standard way that classification are evaluated, though the use of a confusion matrix, along with metrics such as ROC or Kappa. I ended up knowing the overall accuracy, but nothing more. This is not all that useful.
In addition to the content issues described above, the presentation of the manuscript, especially grammar, sentence structure, and word usage, is very problematic. Unfortunately, these problems are so frequent, widespread, and distributed throughout the paper that it's simply not feasible to list them all. Here's a couple of examples from the first paragraph of the introduction:
"It has strong carbon sequestration capacity, 37 promotes the collection of atmospheric dioxide into seawater, and is one of the important 38 contributors of "blue carbon"[4,5]."
"Intergovernmental Panel on Climate Change (IPCC) will issue a landmark report 46 within the range of 1.5°C."
Both of these are sentence fragments. The first one contains an apparent usage error (dioxide should probably be carbon dioxide?) . The second one lack an article (should THE Intergovernmental Panel...) and doesn't really make it clear what the 1.5 degree change refers to. Of course, we know it's global temperature increase, but the grammar here is still incorrect. There are many, many more examples, and unfortunately they do make this manuscript hard to follow, at least for this reader. I would strongly suggest that a revised version of this paper should be written in cooperation with a fully fluent English speaker, who can help the authors avoid simple but significant grammar and word usage errors. This would make for a much better and easier to read paper.
Author Response

(The authors gave the same response as above.)

Round 2
Reviewer 2 Report
The revised paper has addressed all my previous comments, and I suggest to ACCEPT the paper as it is now.
Reviewer 3 Report
The article is much improved and my concerns were addressed adequately.